# Plant roots affect free-living diazotroph communities in temperate grassland soils despite decades of fertilization
Marlies Dietrich[1], Christopher Panhölzl[1], Roey Angel [1,5], Andrew T. Giguere[1], Dania Randi[1], Bela Hausmann [2,3], Craig W. Herbold [1,6], Erich M. Pötsch [4], Andreas Schaumberger [4], Stephanie A. Eichorst [1] & Dagmar Woebken [1] ✉

Fixation of atmospheric $N_2$ by free-living diazotrophs accounts for an important proportion of nitrogen naturally introduced to temperate grasslands. The effect of plants or fertilization on the general microbial community has been extensively studied, yet an understanding of the potential combinatorial effects on the community structure and activity of free-living diazotrophs is lacking. In this study we provide a multilevel assessment of the single and interactive effects of different long-term fertilization treatments, plant species and vicinity to roots on the free-living diazotroph community in relation to the general microbial community in grassland soils. We sequenced the dinitrogenase reductase (*nifH*) and the 16S rRNA genes of bulk soil and root-associated compartments (rhizosphere soil, rhizoplane and root) of two grass species (*Arrhenatherum elatius* and *Anthoxanthum odoratum*) and two herb species (*Galium album* and *Plantago lanceolata*) growing in Austrian grassland soils treated with different fertilizers (N, P, NPK) since 1960. Overall, fertilization has the strongest effect on the diazotroph and general microbial community structure, however with vicinity to the root, the plant effect increases. Despite the long-term fertilization, plants strongly influence the diazotroph communities emphasizing the complexity of soil microbial communities' responses to changing nutrient conditions in temperate grasslands.

Nitrogen (N) availability is crucial for plant growth and limits primary production in terrestrial ecosystems[1]. Through biological nitrogen fixation, specialized microorganisms (diazotrophs) reduce inert atmospheric $N_2$ gas to biologically available ammonia ($NH_3$), which can be readily used by other microorganisms and plants[2,3]. This energy-costly process can represent an important N source and is carried out by symbiotic or free-living diazotrophs. Free-living diazotrophs are ubiquitous in terrestrial ecosystems[4–7], and in temperate grasslands, it has been estimated that they contribute up to 21 kg N ha$^{-1}$ year$^{-1}$[8]. Thus, free-living diazotrophs account for a significant proportion of naturally introduced N into terrestrial ecosystems that lack symbiotic diazotrophs[2,8].

Soil microorganisms are often limited in readily available carbon (C) sources. This is particularly the case for free-living diazotrophs due to the high energy demands of $N_2$ fixation. By releasing large amounts of rhizo-deposits, such as root exudates into the rhizosphere (the narrow zone of soil surrounding roots and influenced by roots), plant roots provide C sources that can potentially meet those high energy demands[9]. Thus, the rhizosphere can provide an attractive habitat for free-living diazotrophs. By creating nutrient-rich microenvironments for microbes, roots are the main drivers of the differentiation of the belowground plant-associated microbiome from bulk soil[10,11]. The rhizosphere is known as a hotspot for microbial colonization[12,13] and activity[14,15], and thus typically exhibits high microbial density. Rhizodeposition further affects soil chemistry by altering the pH, enables the mobilization and acquisition of rather insoluble soil minerals, and contains specific signaling molecules and stimulatory compounds for microbes, but also antimicrobials, that have a selective effect on the soil

[1]Department of Microbiology and Ecosystem Science, Centre for Microbiology and Environmental Systems Science, University of Vienna, Vienna, Austria. [2]Joint Microbiome Facility of the Medical University of Vienna and the University of Vienna, Vienna, Austria. [3]Division of Clinical Microbiology, Department of Laboratory Medicine, Medical University of Vienna, Vienna, Austria. [4]Institute of Plant Production and Cultural Landscape, Agricultural Research and Education Centre, Raumberg-Gumpenstein, Austria. [5]Present address: Institute of Soil Biology and Biogeochemistry, Biology Centre CAS, České Budějovice, Czechia. [6]Present address: Te Kura Pūtaiao Koiora, School of Biological Sciences, Te Whare Wānanga o Waitaha, University of Canterbury, Christchurch, New Zealand. ✉e-mail: dagmar.woebken@univie.ac.at

microorganisms[16]. The rhizoplane describes the root surface, including tightly adhered microbes and the root endosphere (referred to as root in this paper) specifies the inner root tissues inhabited by microbes[15]. Different plant species exude diverse root exudates[17] that support the development of plant-specific rhizosphere microbiomes[12,18,19]. Moreover, different plant species can affect the diversity and composition of diazotroph communities[20–24]. Beneficial rhizosphere-associated bacteria (i.e., plant growth-promoting bacteria) support plant growth and health and can be introduced in agricultural systems to increase crop production[25]. Diazotrophs in the rhizosphere are of particular interest, as they can increase biologically available N and thereby have the potential to reduce the use of chemical N fertilizers in soils[19].

As microbial communities in the rhizosphere are recruited from the bulk soil[26], soil – besides plant species – acts as another important factor shaping plant-associated microbiomes[19,27,28]. Almost all grassland soils throughout Europe are modified by anthropogenic activities[29] such as fertilization, a common practice in agroecosystems to improve soil fertility[30]. The exogenous addition of nutrients can have direct impacts on soil microbial communities or indirect effects by changing soil and plant conditions[30–33]. Similarly, free-living diazotrophs are controlled by fertilization, especially by N and phosphorus (P) inputs. Nitrogen supply via fertilizers mostly downregulate $N_2$ fixation activity, as its uptake is preferred to fixing $N_2$ and abundantly available ammonium can inhibit nitrogenase synthesis[34–36]. In contrast, external P input can enhance $N_2$ fixation rates by alleviating the P limitation of diazotroph communities[37,38]. Fertilization can also affect diazotroph abundance and diversity in soils[39–43], although recently, diazotroph communities in grasslands across four continents were shown to be resilient to short-term fertilization[44]. However, investigations on the effect of long-term nutrient supply on diazotrophic communities are still lacking.

The soils' inherent microbial community as a reservoir for rhizosphere microorganisms[26] as well as plant species as factors shaping the general rhizosphere microbiome have been investigated in the last years[18,28,45] with contrasting reports on whether soil or plant species is the dominant factor shaping the community composition[14,26,46,47]. Similarly, diazotrophic communities are affected by both soil conditions (i.e., different fertilization regimes)[39,40,42,43] and plant species[20]. However, we still lack detailed information on the plant species effect in increasing proximity to the root (i.e., plant root-associated compartments rhizosphere, rhizoplane, root), the effect of fertilizer treatments on diazotroph communities inhabiting these specific root-associated niches and the combinatorial effect of these factors. Here, we provide a multilevel perspective on diazotrophs associated with diverse root-associated compartments of selected perennial plant species (grass and herb species native to temperate grasslands in central Europe) grown in long-term managed grassland soils receiving different levels of nutrient supply (fertilization). Our objective was to determine the single and interactive effects of fertilization, plant species, and microenvironment (root-associated compartments as well as bulk soil) on the free-living diazotroph community composition, diversity, and activity.

In this study, we collected bulk soil and root-associated compartments (rhizosphere soil, rhizoplane, and root) of two grass species (*Arrhenatherum elatius* and *Anthoxanthum odoratum*) and two herb species (*Galium album* and *Plantago lanceolata*) growing in permanent grassland soil treated with different levels of nutrient supply since 1960 (unfertilized, N, P, NPK fertilization) and sequenced the dinitrogenase reductase (*nifH*) gene and the 16S rRNA gene. By combining diazotroph community composition, diversity, and abundance analysis with $N_2$ fixation activity measurements, this research provides insights into the responses of free-living diazotroph communities to fertilization treatment, plant species, and microenvironments in temperate permanent grasslands and brings it in context with the response of the general microbial communities. We hypothesize that all three factors (fertilization, plant species, and microenvironment) shape the diazotrophic community, but that long-term fertilization will have the strongest impact on the diazotrophic community composition. We further hypothesize that with increased vicinity to the root, the effect of the plant species on the diazotroph community composition will increase and potentially surpass the fertilization effect.

## Results

We investigated the diazotroph and general microbial community composition associated with different microenvironments (bulk soil, and root-associated compartments rhizosphere, rhizoplane, root) of two grass species (*Anthoxanthum odoratum* and *Arrhenatherum elatius*) and two herb species (*Galium album* and *Plantago lanceolata*) grown in temperate grassland soils that received mineral fertilizers since 1960. Soil parameters for our investigated fertilization treatments (unfertilized, N fertilization, P fertilization, NPK fertilization) and plant growth data are depicted in Supplementary Table 1. Total N ranged from 0.36 to 0.48% and C:N from 10.81 to 12.18 across these treatments (Supplementary Table 1a). Plants of the two species included in this study were derived from the same plots within each long-term nutrient supply.

### Plant type influences root-associated diazotroph communities

We explored the potential effects of plant type (e.g., grasses or herbs) and plant species on the diazotrophic and general microbial community composition across the root-associated compartments (rhizosphere, rhizoplane, and root together) in all fertilization treatments (Supplementary Table 2). Overall, the plant type (grasses and herbs) had a significant influence on both the diazotrophic (*nifH*: PERMANOVA, $R^2 = 0.011$, $p = 7e-04$) and general microbial community (16S rRNA: PERMANOVA, $R^2 = 0.028$, $p = 1e-04$) across the root-associated compartments in all fertilization treatments (Supplementary Table 2). When parsing the data further into plant species per plant group (namely *Anthoxanthum odoratum* and *Arrhenatherum elatius* as grasses and *Galium album* and *Plantago lanceolata* as herbs), the root-associated diazotroph communities differed significantly between the plant species in the N-fertilized (grasses: PERMANOVA, $R^2 = 0.049$, $p = 0.0473$; herbs: PERMANOVA, $R^2 = 0.081$, $p = 0.0082$) and NPK-fertilized (grasses: PERMANOVA, $R^2 = 0.058$, $p = 0.0321$; herbs: PERMANOVA, $R^2 = 0.091$, $p = 8e-04$) treatments across all root-associated compartments. In addition, the diazotroph community associated with the two herb species also differed in the unfertilized treatment (PERMANOVA, $R^2 = 0.058$, $p = 0.028$) (Supplementary Table 2). The general microbial communities of the three root-associated compartments showed a slightly different pattern. Significant differences were observed between the two herb species in all treatments (Supplementary Table 2) and between the two grass species in the N (PERMANOVA, $R^2 = 0.067$, $p = 0.0114$), unfertilized (PERMANOVA, $R^2 = 0.070$, $p = 0.0086$) and P (PERMANOVA, $R^2 = 0.076$, $p = 0.0046$) treatments (Supplementary Table 2).

Furthermore, we explored if the diazotroph and general microbial community composition differed in each individual compartment (rhizosphere, rhizoplane or root separately) between the two grass species and the two herb species (Supplementary Table 3). Across all fertilization treatments, the diazotroph communities did not differ in any investigated compartment between both grass species or both herb species. However, the general microbial community differed significantly between the herb species in the roots across all fertilization treatments ($R^2 = 0.269$ to $0.351$, $p = 0.006$ to $0.04$) (Supplementary Table 3). Two additional significant differences were observed: the general microbial community composition differed between both grass species only in roots of the N fertilization treatment and between the herb species in the rhizosphere of the NPK fertilization treatment (Supplementary Table 3).

### Vicinity to roots leads to differences in diazotroph community composition in grasses and herbs

The diazotroph community (Fig. 1a) and general microbial community (Fig. 1b) composition associated with the grasses and herbs differed significantly among microenvironments (bulk soil, rhizosphere, rhizoplane and root) across all treatments. Generally, the biggest differences were observed between bulk soil and root communities, whereas rhizosphere

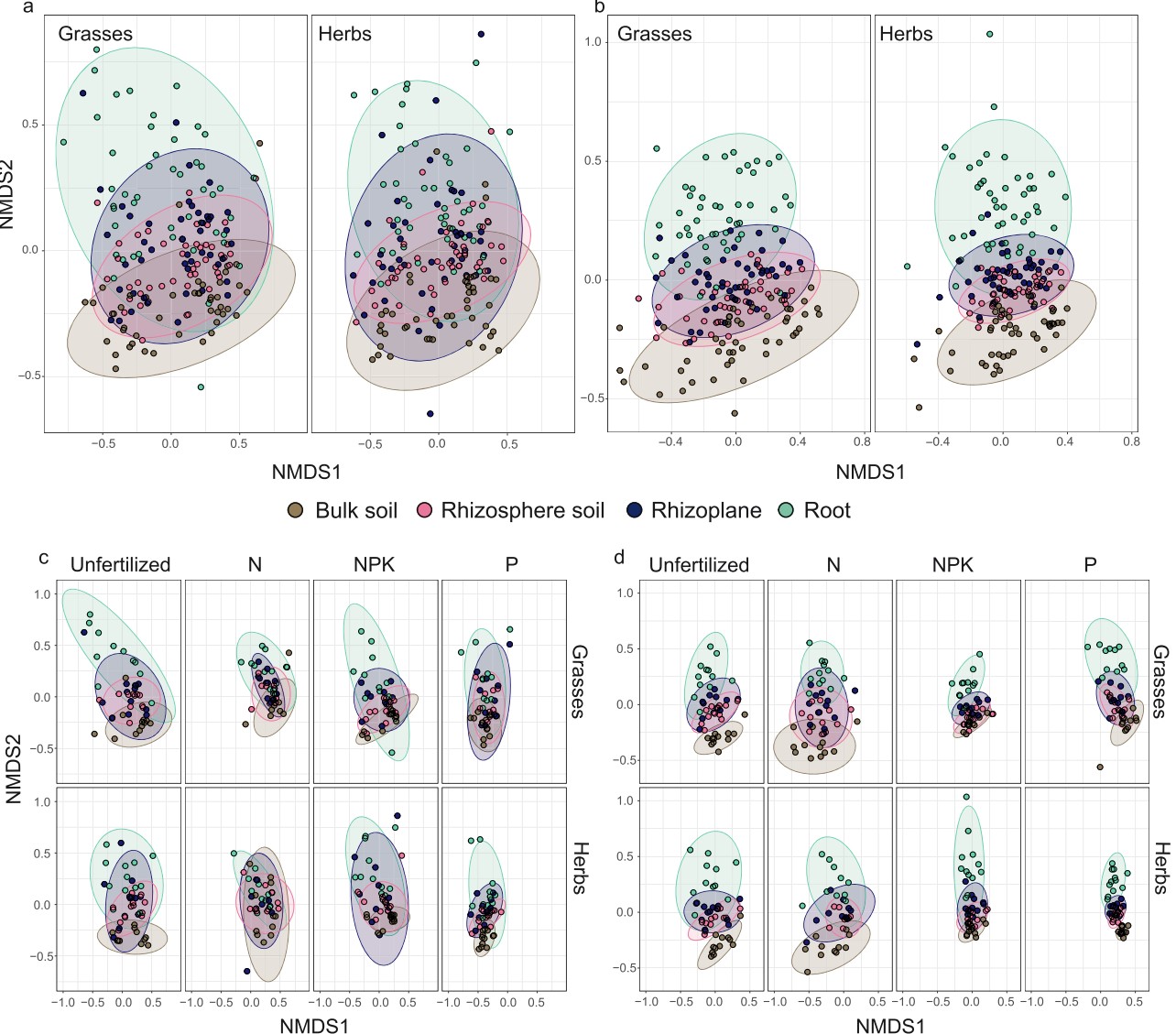

**Fig. 1 | Community composition in microenvironments of investigated plants.**
Non-metric multidimensional scaling (NMDS) ordination plots of **a, c** diazotroph
communities (based on *nifH* genes) (stress value = 0.19) and **b, d** general microbial
communities (based on 16S rRNA genes) (stress value = 0.16) based on Bray–Curtis
metric, illustrating beta diversity of samples obtained from bulk soil, rhizosphere,
rhizoplane, and root. Panels **a** and **b** depict investigated grasses and herbs across all
fertilization treatments, and panels **c** and **d** in each fertilization treatment separately.
Ellipses indicate the 95% confidence interval.

communities did not significantly differ from rhizoplane communities
(Fig. 1a, b, Supplementary Table 4). Overall, a shift in the taxonomic
composition of the diazotroph community from bulk soil to root was
observed (Supplementary Fig. 1). Broadly, members of Alphaproteobacteria
(Rhizobiales) and Cyanophyceae (Nostocales) increased and members of
Deltaproteobacteria (Desulforomonadales) decreased in relative abun-
dances along this continuum in both plant species groups across all ferti-
lization treatments (Supplementary Fig. 1).

Diazotroph and general microbial communities associated with grasses
showed the same significant patterns in pairwise microenvironment com-
parisons across all treatments (Supplementary Table 4). Specifically, bulk
soil communities associated with both grass species differed significantly
from plant-associated compartments (rhizosphere, rhizoplane, and root) in
the unfertilized, P- and N-fertilized treatment, as well as rhizosphere
communities from root communities (Fig. 1c, d, Supplementary Table 4). In
the NPK fertilization treatment, root communities differed from bulk,
rhizosphere, and rhizoplane communities (Fig. 1c, d; Supplementary
Table 4).

Diazotroph and general microbial communities associated with herbs
showed different significance patterns in pairwise microenvironment
comparisons across treatments. Diazotroph communities associated with
bulk soil in both herb species differed significantly from all plant-associated
compartments in the unfertilized treatment and P-fertilized treatment, as
well as rhizosphere diazotroph communities from root diazotroph com-
munities (Fig. 1c, Supplementary Table 4). With NPK fertilization, bulk soil
diazotroph communities differed from herb roots and rhizoplane, while
with N fertilization bulk soil diazotroph communities only differed from
root diazotroph communities (Fig. 1c, Supplementary Table 4). In contrast,
the general microbial communities associated with both herb species dif-
fered significantly in their composition among all microenvironments
except between rhizosphere and rhizoplane across all treatments (Fig. 1b, d;
Supplementary Table 4).

Comparing individual microenvironments between groups of plant
species (grasses and herbs) revealed significant differences in the diazotroph
root and rhizoplane-associated community in the N-fertilized treatment
(Supplementary Table 5). Contrarily, the general microbial community

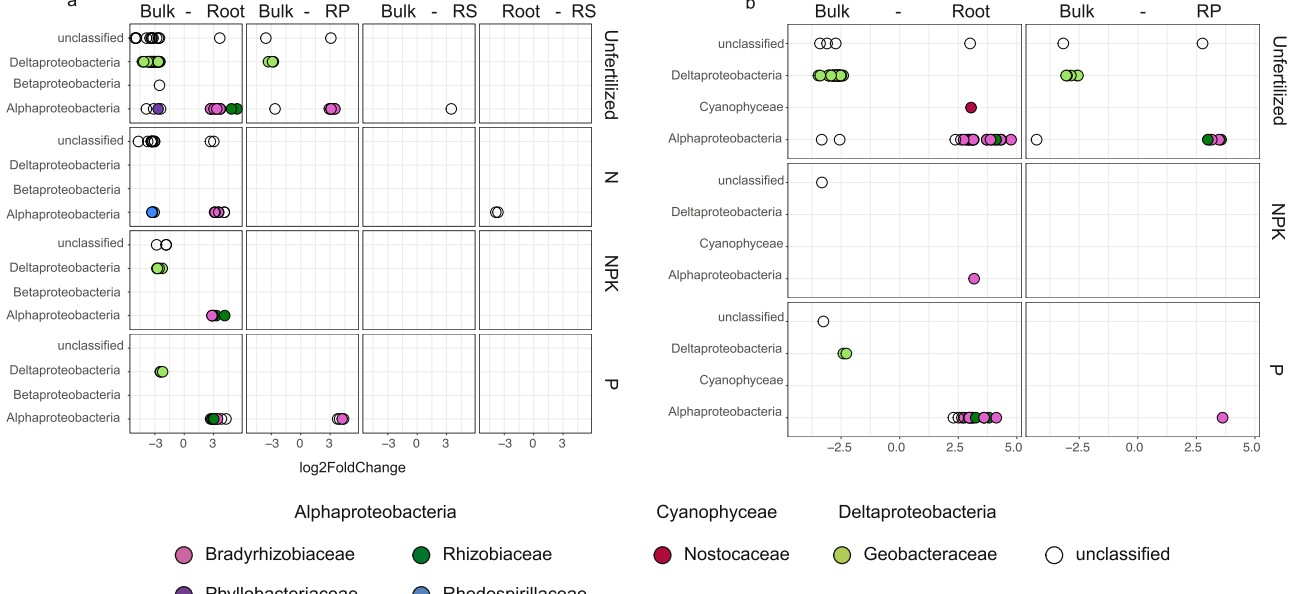

**Fig. 2 | Differential abundance analysis (DESeq2) of diazotrophs between microenvironments.** DESeq2 results showing significantly ($p < 0.05$) differentially abundant *nifH* OTUs between investigated microenvironments associated to **a** grasses and **b** herbs across fertilization treatments. Pairwise microenvironment comparisons are indicated on top of each panel, and log2 fold-change values (*x*-axis) are shown. Dots, colored by family, represent significantly differentially abundant OTUs. Bulk bulk soil, RS rhizosphere soil, RP rhizoplane.

significantly differed between plant types in microenvironments in all fertilization treatments (Supplementary Table 5).

## Plant root vicinity selected for distinct, less diverse diazotroph communities

Overall, the comparison between bulk soil and root displayed the strongest differences in both diazotroph and general microbial community composition across plant type and fertilization treatments (Supplementary Table 4). Diazotroph diversity (Supplementary Fig. 2) significantly decreased with vicinity to the root for both grass and herb species (Dunn Test, Kruskal Wallis multiple comparison, $p = 1.002\mathrm{e}{-}14$). Differential abundance analysis revealed details on OTU-level differences between pairwise microenvironment comparisons in grasses and herbs across nutrient supplies. Grasses showed 124 significantly differentially abundant diazotroph OTUs across microenvironments in all treatments (Fig. 2a, Supplementary Data 1.1). In contrast, we found 94 significantly differentially abundant OTUs across microenvironments associated with herbs (Fig. 2b, Supplementary Data 1.2) in the unfertilized, NPK and P fertilization treatments.

The unfertilized treatment harbored many of the significantly different diazotrophic OTUs, specifically between bulk soil and root in grasses (59 out of 105 OTUs, Supplementary Data 1.1) and herbs (61 out of 80 OTUs, Supplementary Data 1.2) (Fig. 2a, b). Differential abundance analysis among microenvironments revealed large differences within unclassified OTUs (Fig. 2, Supplementary Data 1.1 and 1.2). Within the classified community, significantly more abundant Geobacteraceae and Rhodospirillales OTUs were present in bulk soil compared to the root in grasses and herbs. In contrast, Rhizobiales, Bradyrhizobiales, and Nostocales OTUs were significantly more abundant in the root versus the bulk soil environment (Fig. 2, Supplementary Data 1.1 and 1.2). Significantly more abundant OTUs of Rhizobiales and Bradyrhizobiales were also found in the rhizoplane compared to bulk soil in the unfertilized and P-fertilized treatment in both plant species groups (Fig. 2, Supplementary Data 1.1 and 1.2).

Diazotroph abundance in the rhizosphere of investigated plants (Fig. 3) was significantly lower compared to bulk soil abundances in unfertilized plots (ANOVA, $p = 3.25\mathrm{e}{-}04$). No difference was observed between

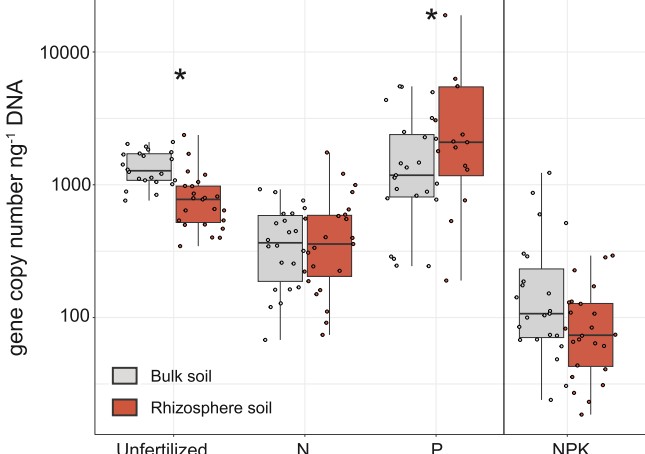

**Fig. 3 | Diazotroph abundances across fertilization treatments.** Abundances obtained by qPCR quantification of *nifH* gene in bulk (gray) and rhizosphere soil (red) of all investigated plants across different fertilization treatments. Unfertilized, N and P field treatments were sampled in 2014, and NPK treatment was sampled in 2018. The upper and lower hinges indicate the first and third quartiles. Significant differences ($p < 0.05$) based on ANOVA results between both microenvironments are indicated (*). Exact sample sizes: unfertilized ($n = 45$ biologically independent samples), N ($n = 45$ biologically independent samples), NPK ($n = 47$ biologically independent samples), P ($n = 44$ biologically independent samples).

diazotroph abundance in the rhizosphere and bulk soil of N- and NPK-fertilized soils. Only in the P addition treatment, especially with the herb *Galium album*, diazotroph abundance was significantly higher in rhizosphere soil than in bulk soil (ANOVA, $p = 0.019$) (Supplementary Fig. 3). Although the NPK fertilization treatment appears to have an overall lower diazotroph abundance, these samples were collected at a later sampling year as compared to the unfertilized, N and P fertilization treatments (for more details, see Material and methods).

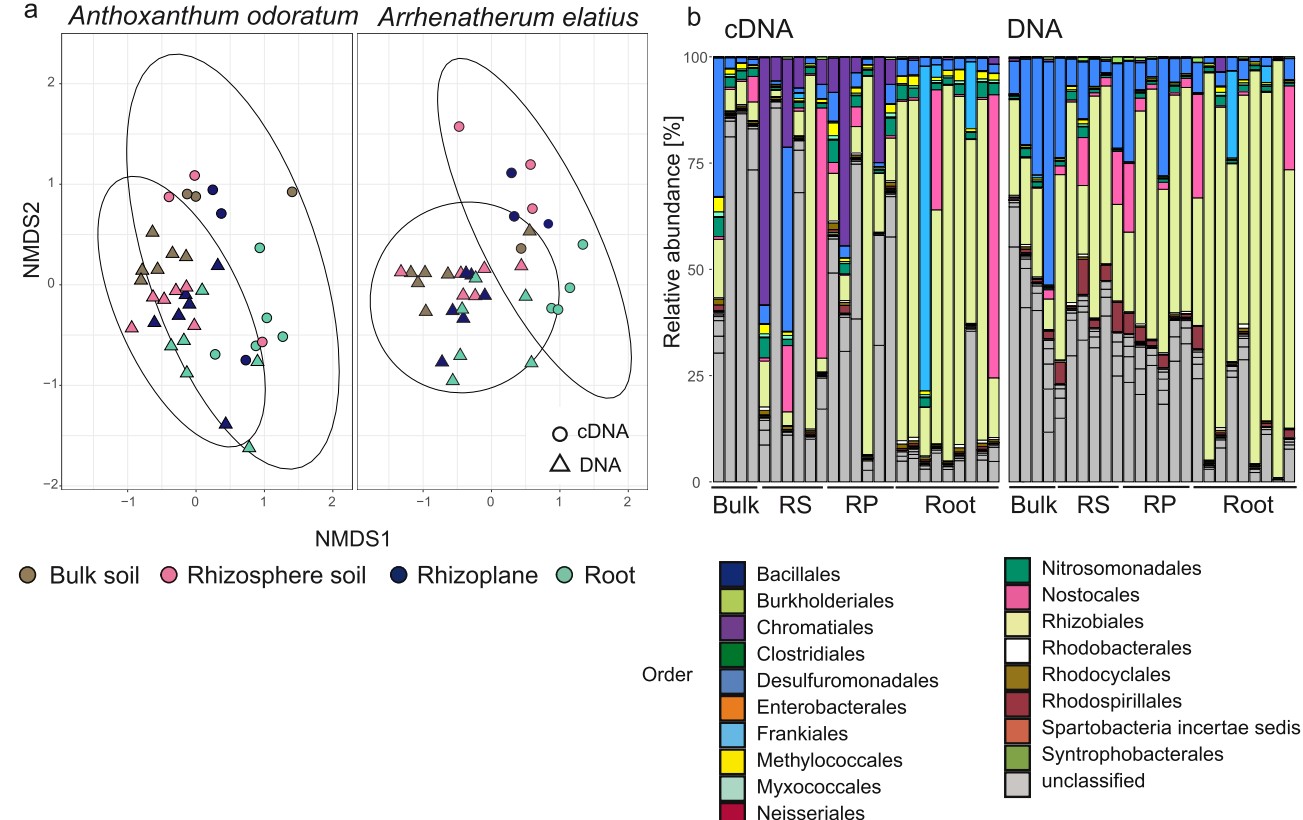

**Fig. 4 | Composition of active (cDNA-based) and DNA-based diazotroph communities associated with grasses. a** Non-metric multidimensional scaling (NMDS) ordination plots of *nifH* genes and transcripts (cDNA) in bulk soil, rhizosphere soil, rhizoplane, and roots of the grass species *Anthoxanthum odoratum* and *Arrhenatherum elatius* in the unfertilized treatment based on Bray–Curtis metric (stress value = 0.20). Ellipses indicate the 95% confidence interval. Exact sample sizes cDNA: bulk soil (*n* = 4 biologically independent samples), rhizosphere (*n* = 6 biologically independent samples), rhizoplane (*n* = 6 biologically independent samples), root (*n* = 9 biologically independent samples. Exact sample sizes DNA: for bulk soil, rhizosphere, rhizoplane, root (*n* = 12 biologically independent samples). **b** Taxonomic composition of the active diazotroph community (cDNA) and the DNA-based diazotroph community associated with bulk soil, rhizosphere, rhizoplane and roots of both grasses of the unfertilized treatment. Stacked bars reflect relative abundances (%) and are colored based on order level. Exact sample sizes cDNA and DNA: bulk soil (*n* = 4 biologically independent samples), rhizosphere (*n* = 6 biologically independent samples), rhizoplane (*n* = 6 biologically independent samples), root (*n* = 9 biologically independent samples). Bulk bulk soil, RS rhizosphere soil, RP rhizoplane.

### Actively transcribing diazotrophs across microenvironments constituted a distinct subset of the community

To assess actively transcribing diazotrophs, samples from all microenvironments from both grass species of the unfertilized plots were used. The community composition of actively transcribing diazotrophs (based on *nifH* cDNA analysis) in microenvironments of both grass species of the unfertilized plots significantly differed among the two investigated grass species (PERMANOVA, $R^2 = 0.034$, $p = 0.0053$) and from the DNA-based diazotroph community (PERMANOVA, $R^2 = 0.105$, $p = 0.0001$, Fig. 4). In each microenvironment, about 50% of the OTUs of the DNA-based community were also found in the active community with the highest proportion of active OTUs in the root (bulk soil: 41%, rhizosphere: 56%, rhizoplane: 45%, root: 60%). Among the microenvironments, the community composition of the active diazotrophs differed significantly between bulk soil and root (PERMANOVA, $R^2 = 0.202$, $p = 0.0192$), and rhizoplane and root (PERMANOVA, $R^2 = 0.164$, $p = 0.0492$), whereas the DNA-based community composition exhibited significant differences only between bulk soil and root (PERMANOVA, $R^2 = 0.194$, $p = 0.0252$).

The relative abundance of actively transcribing members of Chromatiales, Nitrosomonadales and Methylococcales was higher in all microenvironments, except the root compared to the DNA based community. Conversely, the relative abundance of actively transcribing members of Nostocales and Frankiales was higher in the root than in the other microenvironments (Fig. 4b).

### N fertilization reduced diazotroph diversity and abundance and selected distinct community members in bulk soils

The long-term fertilization regime selected for distinct diazotroph (PERMANOVA, $R^2 = 0.37$, $p = 1e{-}04$) and general microbial communities in bulk soils (PERMANOVA, $R^2 = 0.283$, $p = 1e{-}04$) (Fig. 5a, b). P-fertilized soils exhibited the highest diazotroph diversity (Supplementary Fig. 4a) and abundances (Fig. 3), significantly different from less diverse and less abundant diazotroph communities in unfertilized (Dunn test, $p = 2.545e{-}04$), NPK-fertilized (Dunn test, $p = 2.455e{-}05$) and N-fertilized (Dunn test, $p = 1.555e{-}06$) plots. However, the dispersion between groups was not homogenous (ANOVA, $p = 0.02$), most likely due to the high group variance dispersion in N-treated samples. Dispersion was homogenous when N treatment was excluded from analysis (ANOVA, $p = 0.3125$). Plots of three of the investigated treatments (unfertilized, N-fertilized, and P-fertilized) were sampled in 2014, while NPK plots were sampled in 2018. However, we compared bulk soil samples that were derived from the same plots in both years (2014 and 2018) to investigate the possibility for changes in the community due to time. Our analysis demonstrates that in bulk soils, neither the diazotrophic (Supplementary Fig. 5a) nor the general microbial community (Supplementary Fig. 5b) changed significantly in the investigated years in the N, P, and unfertilized treatments (based on PERMANOVA analysis, see text to Supplementary Fig. 5 for more information). Therefore, we felt confident to combine the data from 2014 and 2018 to explore the effect of fertilization on the diazotrophic community.

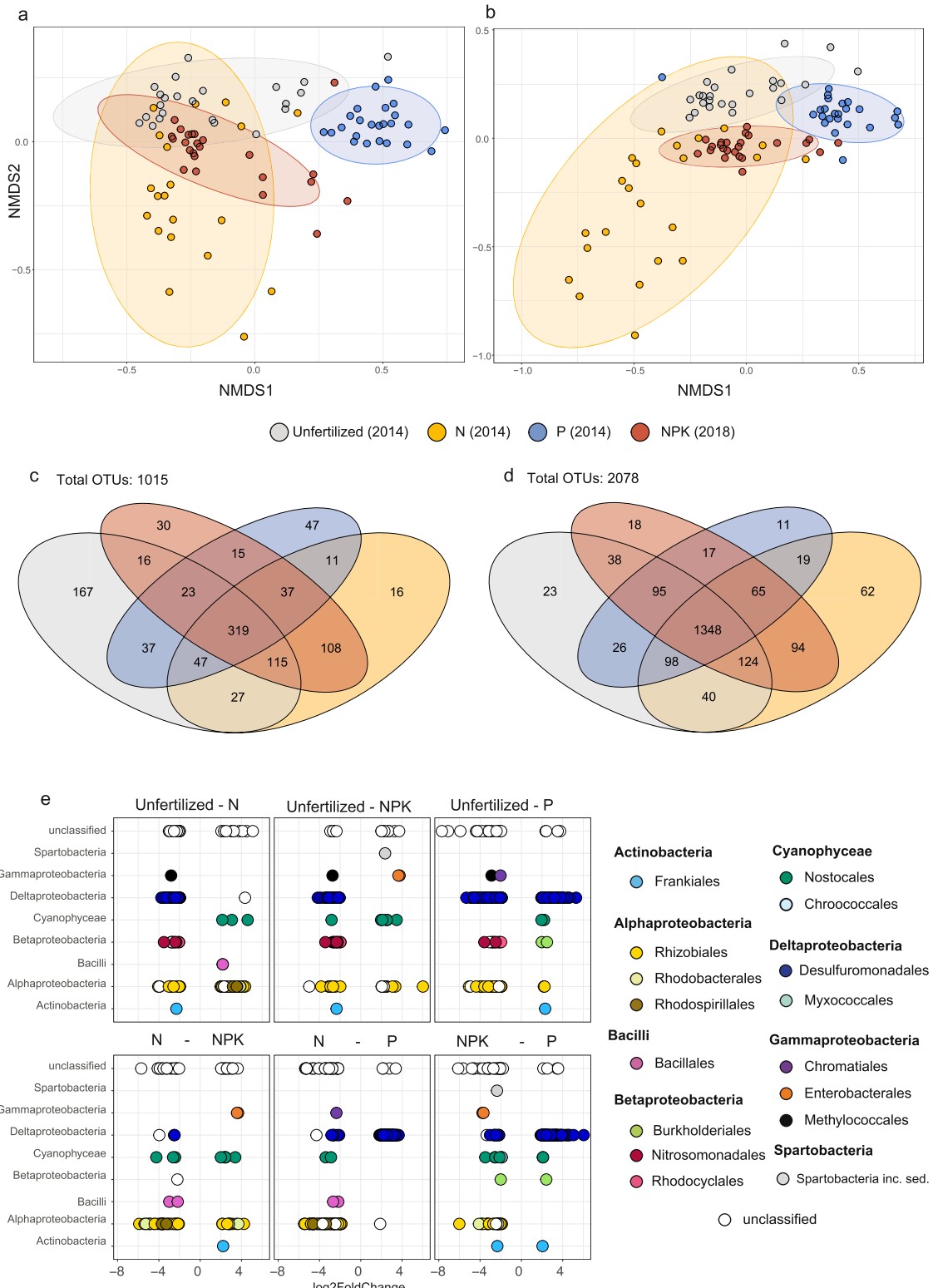

**Fig. 5 | Community composition and differential abundance analysis of microbial communities in bulk soil across fertilization treatments.** Non-metric multidimensional scaling (NMDS) ordination plots of **a** diazotroph communities (based on *nifH* genes) (stress value = 0.12) and **b** general microbial communities (based on 16S rRNA genes) (stress value = 0.09) in bulk soil from all investigated fertilization treatments (unfertilized—gray; N—yellow, NPK—red; P—blue) based on Bray–Curtis metric. Years correspond to the sampling timepoints of the respective soil (unfertilized—2014, N—2014, P—2014, NPK—2018). Ellipses indicate the 95% confidence interval. Venn diagrams depict shared and unique OTUs of sequenced **c** diazotroph and **d** general microbial communities across investigated fertilization treatments. **e** Differential abundance analysis (DESeq2) results showing significantly ($p < 0.05$) differentially abundant *nifH* OTUs in bulk soils across investigated fertilization treatments. Pairwise treatment comparisons are indicated on top of each panel, and log2 fold-change values (*x*-axis) are shown. Dots, colored by order, represent significantly differentially abundant OTUs.

**Fig. 6 | N₂ fixation potential of diazotrophs across unfertilized, N- and P-fertilized treatments.** $N_2$ fixation potential was obtained upon incubation with root exudates (blue) or no carbon addition (white) for 3, 7, and 21 days. $^{15}N$ isotopic content (atom%) is shown; the dashed black line represents the natural abundance control (average value of multiple measurements), lower and upper hinges represent the first and third quartiles. Significant differences ($p < 0.05$) to $^{14}N$ control (ANOVA) are indicated (*). Exact sample sizes per day and root exudate addition or no carbon addition: Unfertilized ($n = 3$ biologically independent samples), N ($n = 3$ biologically independent samples), P ($n = 3$ biologically independent samples).

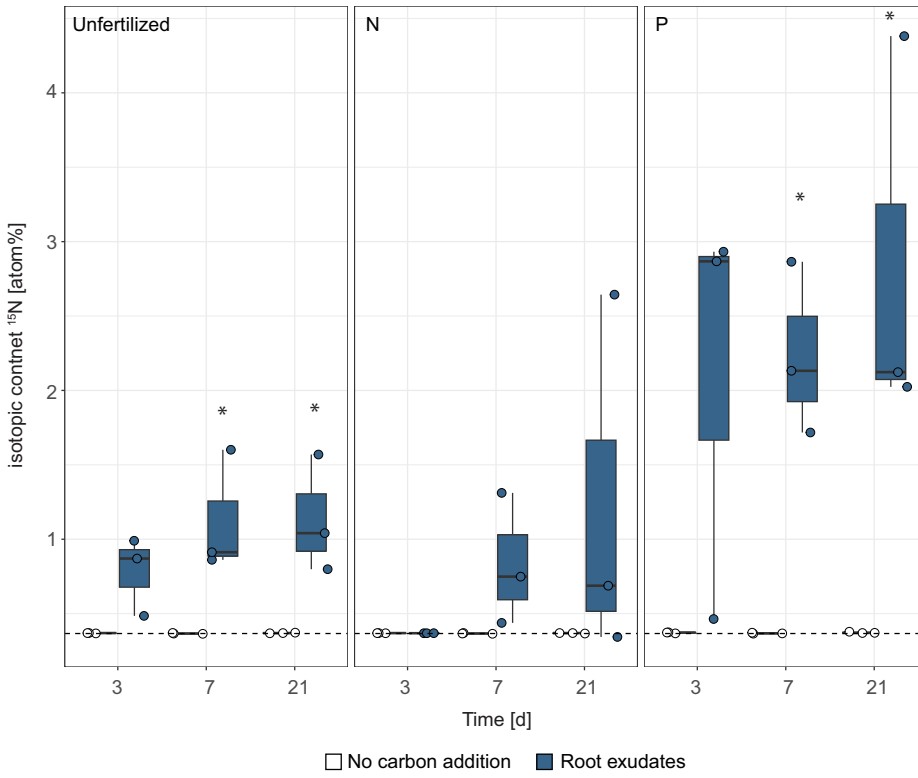

Only 31% (319 OTUs) of all diazotroph OTUs (Fig. 5c) in bulk soils were shared among all treatments, compared to 64% (1348 OTUs) of shared OTUs in the general soil microbial community (Fig. 5d). Interestingly, most of the unique diazotroph OTUs were found in unfertilized (167 OTUs, 16% of all OTUs) and P-fertilized (47 OTUs, 4.7% of all OTUs) plots (Fig. 5c). Soils from N and NPK treatments shared more OTUs (*nifH*: 108 OTUs, 16S rRNA: 94 OTUs) than soils from unfertilized and P-fertilized treatments (*nifH*: 37 OTUs, 16S rRNA: 26 OTUs). Changes in the relative abundance of members of the diazotroph community were observed across the different fertilization treatments (Supplementary Fig. 1). More specifically, the unfertilized and P treatment showed an increase in significantly differentially abundant diazotroph OTUs belonging to Desulfuromonadales compared to N-fertilized and fully fertilized soils, which harbored more significantly differentially abundant diazotroph OTUs belonging to Rhizobiales, Rhodobacterales and Rhodospirillales (Fig. 5e, Supplementary Data 1.3). The unfertilized treatment was characterized by the presence of significantly differentially abundant diazotroph OTU_kh3_m2n (Methylococcales) and diazotroph OTUs belonging to Nitrosomondales and Rhodocyclales (OTU_d8a_qub, OTU_bns_q4q, OTU_io9_vak, OTU_959_vcr, OTU_c5f_i14, OTU_ryz_ypn, OTU_jdl_fgk) that were absent in all other fertilization treatments (Fig. 5e, Supplementary Data 1.3). Among all pairwise comparisons, significantly differentially abundant diazotroph OTUs of Enterobacterales were exclusively present in NPK-treated soils, significantly differentially abundant OTUs of Bacillales (OTU_8-qe_1lv, OTU_lrv_qrl) exclusively found in N-treated soils and significantly differentially abundant OTUs of Frankiales (OTU_rzx_i75 and OTU_id6_sni) only present in P-treated soils (Fig. 5e, Supplementary Data 1.3).

**Unfertilized and P-fertilized soils exhibit increased N₂ fixation potential**

To assess the potential for $N_2$ fixation in the differently fertilized soils, we mimicked root exudation by supplying easily accessible carbon sources (artificial root exudates) to soils of unfertilized, N- and P-fertilized treatments and performed $^{15}N_2$ gas $N_2$ fixation assays. In general, P-fertilized, and unfertilized soils exhibited higher $N_2$ fixation activity than N-fertilized soils (Fig. 6). Significant $N_2$ fixation activity was already detected after seven days of incubation in unfertilized (ANOVA, $p = 0.035$) and P-fertilized (ANOVA, $p = 0.005$) soils and increased after 21 days (unfertilized: ANOVA, $p = 0.025$; P fertilization: ANOVA, $p = 0.032$). In contrast, no significant $N_2$ fixation was detected in N-fertilized soils upon root exudate supply or in soils that did not receive external root exudates across all treatments.

**Combinatorial analysis reveals an increasing effect of plants on diazotroph and general microbial community structure within the vicinity of the root**

Overall, fertilization, microenvironments, and plant species had significant effects on the structure of the microbial communities. For both, the diazotroph community and the general microbial community, fertilization (*nifH*: PERMANOVA $R^2 = 0.193$, $p = 0.001$, 16S rRNA: PERMANOVA $R^2 = 0.2$, $p = 0.001$) had a stronger effect than microenvironment (*nifH*: PERMANOVA $R^2 = 0.053$, $p = 0.001$, 16S rRNA: PERMANOVA $R^2 = 0.111$, $p = 0.001$) and plant species (*nifH*: PERMANOVA $R^2 = 0.021$, $p = 0.001$, 16S rRNA: PERMANOVA $R^2 = 0.047$, $p = 0.001$). This effect was also reflected by the number of significantly differentially abundant diazotroph OTUs, which was higher when comparing the fertilization treatments (498 OTUs), than when the investigated microenvironments were compared (147 OTUs). Similarly, also for the general microbial community there were more significantly differentially abundant OTUs among all fertilization treatments (821 OTUs) than among the investigated microenvironments (384 OTUs).

Investigating the effect of fertilization and plant species in each microenvironment individually revealed interesting patterns of microbial community structures. When performing this combinatorial analysis, this pattern was observed independent of whether the NPK treatment was included (Supplementary Table 6) or not (Supplementary Table 7). Across all investigated microenvironments, fertilization had a stronger effect than

**Fig. 7 | Conceptual figure.** Illustration of the influence of fertilization treatment (unfertilized, N, P, NPK) and plant species (*Anthoxanthum odoratum*, *Arrhenatherum elatius*, *Plantago lanceolata*, *Galium album*) on the diazotroph and general microbial community. Arrows represent significant effects based on Bray–Curtis dissimilarity, permutational multivariate analysis of variance (PERMANOVA) of fertilization treatment (purple), and plant species (green) on the microbial community in bulk soil, rhizosphere soil, rhizoplane, and root. The size of the arrow and the numbers above correspond to the variance explained by the respective factor in each investigated microenvironment (e.g., the larger the arrow, the more of the variance is explained, Supplementary Table 6). Note that the influence of the plant increases with the vicinity of the root, especially in the general microbial community.

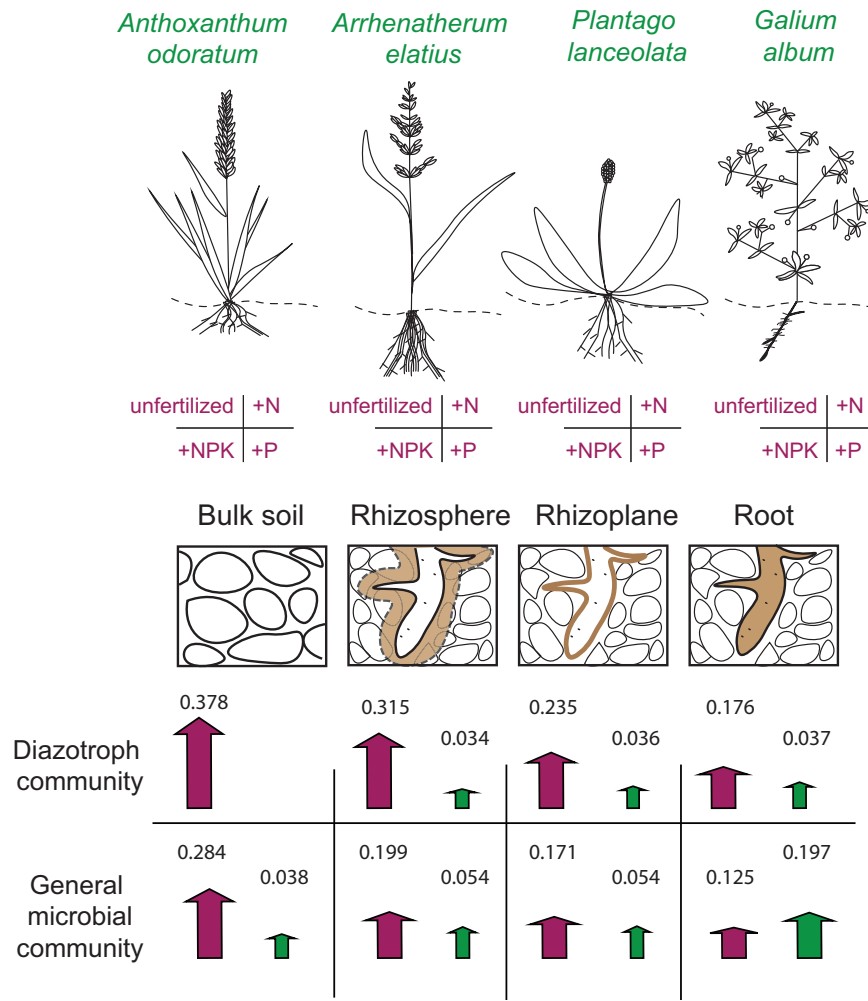

plant species on the diazotrophic community (Fig. 7). Within bulk soil, only fertilization showed a significant effect on diazotroph community structure (PERMANOVA $R^2 = 0.37$, $p = 0.001$). However, with vicinity to root, the effect of plant species increased (Supplementary Table 6). The general microbial community showed a different picture, as both fertilization and plant species, as well as their interactive effects, had significant effects across all microenvironments (Fig. 7, Supplementary Table 6). In bulk soil, rhizosphere, and rhizoplane, fertilization still had the strongest effect on the general microbial community; however, within the root, the effect of the plant species surpassed the influence of the fertilization (Fig. 7, PERMANOVA, $R^2 = 0.19$, p = 1e−04, Supplementary Table 6).

## Discussion

Our study emphasizes the complex responses of free-living diazotrophs and general microbial communities in the soil-plant root continuum to changing nutrient conditions in temperate permanent grassland ecosystems. To the best of our knowledge, this is one of the first investigations that took a combinatorial assessment of the effects of different microenvironments (including bulk soil and various root-associated compartments) of multiple plant species receiving different fertilizer nutrients on the microbial community composition in permanent grassland soils that are vital to European agriculture. We demonstrated that both the nutrient supply and microenvironments shaped the microbial community in the investigated soils. Our data suggest that fertilization treatment had the strongest effect on diazotroph and general microbial community structure, however with

vicinity to the root, the plant-effect increased and even surpassed the treatment effect in the general microbial community.

It has long been established that plant species have a critical impact on soil microbial communities[26,48] and even different plant species growing within the same soil type influence the soil microbial community composition differently[46,49]. In our system, a temperate permanent grassland with a diverse plant community, the investigated perennial grasses and herbs grown in the same soils (continuously fertilized since 1960) assembled distinct plant-associated diazotroph and general microbial communities across all fertilization treatments. Evidence of the specificity of diazotroph communities associated with individual plant species has been shown for the rhizosphere of medicinal plants and wheat[20,21], however, our findings illustrate that grass- and herb-specific diazotroph microbial communities in various root-associated compartments persist in mixed grasslands despite more than five decades of mineral fertilizer applications. Our results demonstrate a considerable change in diazotroph community composition and a reduced diversity from bulk soil to the root. Shifts in diversity and community composition along the soil-root continuum have been discussed extensively for general microbial communities[10,14,15,28,50], but until now have not been shown for diazotroph communities across multiple plant-associated compartments of various plant species from temperate permanent grassland soils. Additionally, our results provide the first insights on shifts in the active (cDNA-based) diazotroph community across specific microenvironments associated with grasses. The changes in microbial diversity and composition from bulk soil to root can be linked to processes

that occur during the root microbiome differentiation that reduce niche dimension with increasing vicinity to the root[15]. Comparisons of bulk soil and root revealed the strongest effect on diazotroph and general microbial communities across all plants and fertilization treatments in our system. Previous studies on rice and *Arabidopsis thaliana* general microbial communities[51–53] reported that microbial communities inhabiting root tissues were strongly differentiated from rhizosphere or bulk soil communities. As suggested in multi-step models for root microbiota differentiation from bulk soil communities[15,28], roots are the driving force in the differentiation of the plant-associated microbiome, as root volume, length, and especially the chemical composition of root exudates varies across individual plant species[26,54]. Given the energetic demand of $N_2$ fixation, the diazotroph guild toward the root was likely shaped by different root exudates of grasses and herbs in the investigated grassland. In support of this conjecture, we observed increased $N_2$ fixation potential when artificial root exudates were supplied to these soils. The significantly differently present diazotrophic taxa found in the root compartment compared to bulk soil, such as members of Rhizobiales and Bradyrhizobiales are related to ubiquitous root communities[55].

In agreement with previous studies, we showed that long-term N fertilization can satisfy N requirements of microorganisms and dramatically reduce $N_2$ fixation activity, diazotroph diversity and abundance[39,40,56,57]. Besides selection pressure or inhibition of the nitrogenase enzyme[34–36], the decrease in diazotroph diversity could be caused by increased soil acidity with external N input. Acidic soils generally exhibit lower bacterial diversity, and diazotrophs especially have a narrow growth tolerance to pH[31]. It has been shown previously that soil pH plays an essential role in diazotroph community assembly in bulk and rhizosphere soils[21]. Indeed, N fertilization led to a drastic decrease in pH (on average 4.4, see Supplementary Table 1) in our investigated permanent grassland. N-supplied soils did not show any significant increase of $N_2$ fixation activity upon simulated root-exudation, implying that the large dose of N fertilizers exceeded the root exudate energy supply[58] or a lack of demand for $N_2$ fixation. The diazotrophic community in soils that received external N was mainly dominated by members of Alphaproteobacteria, especially from the order Rhizobiales and the genus *Bradyrhizobium*, which are known copiotroph organisms and facultative $N_2$ fixers that might benefit from fertilization to support their own growth instead of fixing $N_2$[39,59]. In this study, we added a dataset (NPK fertilization, sampled in 2018) to the larger dataset of samples stemming from 2014 (unfertilized, N and P fertilization). Tests confirmed that the soil microbial community composition of the investigated treatment plots did not change significantly between these different sampling years (Supplementary Fig. 5); however, we cannot fully exclude the possibility of an effect of different sampling times. We addressed this concern as follows: for statistical analysis where we directly combined data from all treatments, we also repeated the analysis excluding the NPK treatment to rule out that the observed effects stemmed from different sampling times. The results with this reduced set of fertilization treatments (N, P, unfertilized, Supplementary Table 7) were virtually identical to the observed effects when including the NPK treatment in the analysis (Supplementary Table 6), supporting the applied approach of merging both datasets.

Phosphorus availability was a clear driver for diazotroph community composition and led to a strongly increased potential for $N_2$ fixation activity in our investigated grassland. Phosphorus is a key nutrient in energy production, and its availability controls highly energy-demanding processes, such as the adenosine triphosphate (ATP) requiring $N_2$ fixation process[60]. Therefore, diazotrophs have high P requirements[61,62], and in our system, diazotrophs seemed to be limited in P as the amendment of P yielded the highest diazotroph diversity, abundance, and highest $N_2$ fixation activity upon root exudates supply. Increased P availability could indirectly affect plant growth by enhancing $N_2$ fixation in soils and thus alleviating N limitations[61]. Moreover, it has been reported that P addition can modify plant root structure[63] leading to longer root formation and that it influences root exudation patterns[64], which might have stimulating effects on the soil microbial community.

Similar to the P-fertilized plots, we observed high diazotroph diversity and abundance in the unfertilized treatment. This could indicate that these diazotrophs are more competitive without N fertilization. Members of the class Deltaproteobacteria, especially the order Desulfuromonadales and family Geobacteraceae, known oligotrophic organisms[39,65], were significantly more abundant in the P and unfertilized treatment. Perhaps these organisms are outcompeted under high N conditions, due to their low capacity of downregulating $N_2$ fixation[39,66].

Our results not only supported the idea that changes in soil nutrient amendments influence the composition and structure of soil diazotrophic and general microbial communities[39,40,42,58,67,68], but revealed that despite the long-term mineral fertilizer application, plants strongly shaped the diazotroph communities. This is in stark contrast to previous studies on long-term fertilized agroecosystems dominated by one single plant species where fertilization generally surpassed the influence of plants on the microbial communities[58,69]. This was possible due to a combinatorial approach using plant type, plant-associated microcompartments, and a site with long-term fertilization regimes since the 1960s. Our data further highlight the need to study individual microenvironments in the soil-plant root continuum to disentangle the influence of the plant on soil microbial community composition and structure, which has implications in both long-term and short-term fertilization treatments. Contrary to other studies, we compared our findings to the general microbial community and found similar patterns illustrating that the influence of nutrient supply and microenvironments extends beyond the diazotrophic guild. Taken together, our results indicate that despite the long-term nutrient supply the community was still influenced by root-derived substrates in the investigated permanent grassland soils. The plant effect increased with vicinity to root, indicating that the root-influenced community might be less susceptible to the effects of long-term fertilization. Overall, our study provides insights in the free-living diazotroph community structure associated with plants grown in long-term fertilized grasslands and yields important information on the impact of agricultural management practices on plant-microbe interactions.

## Material and methods
### Site description
Soil and plant samples were collected from a long-term nutrient deficiency experiment at the Agricultural Research Center Raumberg-Gumpenstein in Styria, Austria (47°29′37″N, 14°06′10″E). The site is located 710 m a.s.l. with a mean annual temperature of 8.5°C and a mean annual precipitation of 1080 mm (observation period 1991–2020). According to the WRB-system[70] (IUSS Working Group WRB, 2015) the soil is classified as Dystric Cambisol (arenic, humic). Established in 1960, the experiment aims to study the effect of mineral nutrient supply on the productivity of a typical meadow plant community. Due to its long-term monitoring, it represents a valuable research site for the study of nutrient impacts on microbial communities in temperate permanent grasslands. Small plots, arranged in a randomized block design receive different combinations of mineral fertilizers. Since 1960, the same fertilization treatments and amounts were continuously applied to the plots. Three randomly placed replicate plots exist per treatment. Plant biomass is mown three times a year and removed. For our study, the following four fertilization treatments were chosen: (1) fully-fertilized plots (hereafter referred to as NPK-fertilized): 120 kg N + 60 kg $P_2O_5$ + 240 kg $K_2O$ ha$^{-1}$ y$^{-1}$; (2) N-fertilized: 120 kg N ha$^{-1}$ y$^{-1}$; (4) P-fertilized: 1 × 120 kg $P_2O_5$ ha$^{-1}$ y$^{-1}$; and (4) unfertilized: no nutrient supply. Edaphic properties and plant growth data of these treatment regimes are summarized in Supplementary Table 1. One-third of the total N rate is applied at the beginning of the growing season, one-third right after the first cut (end of May or early June), and one-third after the second cut (end of July), whereas the full rate of P and K is applied in autumn (mid-October). Nitrogen was applied in the form of ammonium nitrate, P by means of hyperphosphate, and K as potassium chloride. Based on regular vegetation surveys and monitored abundance patterns across our treatments of interest, four representative plant species, two grass species (*Arrhenatherum elatius* (tall oat grass) and *Anthoxanthum odoratum*

(sweet vernal grass)) and two herb species (*Galium album* (white bed-straw) and *Plantago lanceolata* (ribwort plantain)) were selected for sampling and analysis.

## Plant and soil sampling

Each plant species was sampled in duplicates from each of the three replicate treatment plots, adding up to a total of 6 plants per plant species and treatment (Supplementary Table 8). Plants and soil were sampled before the first cut from N-fertilized, P-fertilized and unfertilized treatments in May 2014 (May 20th and 21st), and from NPK-fertilized plots in May 2018 (May 27th and May 28th). No significant changes were detectable in the diazotroph and general microbial community composition in the treatments over time (Supplementary Fig. 5). For statistical analysis where we directly combined data from all treatments, we repeated the analysis excluding the NPK treatment to rule out that the observed effects stemmed from different sampling times. The results indicated that the observed effects of fertilization treatments (N, P, unfertilized alone) were virtually identical to the previously observed effects, where we included the NPK treatment in the analysis (Supplementary Fig. 5, Supplementary Tables 6 and 7). As such, the differences in sampling times did not significantly affect the observed results, justifying the merging of these samples in our analysis. Plants were carefully removed from the soil; any loose soil that was not adhered to the roots upon gentle shaking was defined as bulk soil. For the separating of root-associated habitats we used an adaptation of a previously published protocol[71]. Briefly, the root system was excised from the plant using ethanol-sterilized scissors. The roots were then placed in a sterile, 50 ml tube containing phosphate-buffered saline (PBS) and shaken on a table-top shaker at 200 rpm for 10 min. The roots were removed, and the remaining rhizosphere soil slurry was centrifuged at 1768 rpm for 10 min at room temperature. The resulting soil pellet was transferred to 2 ml tubes and defined as a rhizosphere soil sample. The washed roots were then shaken in a PBS-Tween solution at 200 rpm for 30 min, transferred to a fresh tube, and defined as a root sample. The remaining rhizoplane solution was centrifuged at 1768 rpm for 30 min at room temperature, and the pellet was defined as a rhizoplane sample based on[71]. Additionally, we amended this protocol whereby the rhizoplane supernatant was filtered through 0.2 μm filters mounted on 0.45 μm support filters. The rhizoplane sample thus represents a combination of pellet and filter samples to capture as many rhizoplane cells as possible. All samples were immediately shock-frozen in liquid nitrogen and stored at −80 °C before further processing. Root samples were milled in liquid nitrogen for 1 min; the frozen root powder was stored at −80 °C.

In addition to the plant-derived samples, soil cores (2.5 cm diameter) were taken from the four chosen treatment plots (NPK-, N-, and P-fertilized, and no fertilization). Four to five soil cores (10 cm depth) were randomly sampled from each of the three replicate plots per treatment (yielding ~200 g of soil per plot), homogenized using a 2 mm sieve, and immediately stored at −80 °C or processed for $^{15}N_2$ incubations.

## $^{15}N_2$ incubations with different carbon sources

N$_2$ fixation activity was evaluated using $^{15}N_2$ tracer assays[72]. Soil of the treatments sampled in 2014 (N- and P-fertilized, and unfertilized) was used for the assay upon homogenization with a 2 mm sieve. Approximately, 2.2 g of soil was incubated with a $^{15}N_2$ enriched atmosphere ($^{15}N_2$:He:O$_2$ (40:40:20)) (CAMPRO Scientific, Berlin, Germany) supplemented with two different carbon sources in a muffled 10 ml glass serum bottle, sealed with a butyl-rubber stopper. An artificial liquid root exudate mix (RE; modified from[73] excluding the amino acids) was added to yield 1.8 mg C g$^{−1}$ soil. Molecular-grade water (Roth, Karlsruhe, Germany) was added to the no-carbon control incubations to correct for any effect due to the water content. Each incubation condition was performed in triplicates and natural abundance controls (incubated without the addition of $^{15}N_2$) were included for all conditions. Samples were incubated in the dark for either 3, 7, or 21 days at room temperature. At the respective sampling point, samples were frozen at −80 °C for further analysis.

## Nucleic acid extraction, RNA purification, and cDNA synthesis

Nucleic acids (NA) were extracted from bulk soil (0.4 g) rhizosphere soil (0.2–0.4 g), rhizoplane (0.3–0.4 g), rhizoplane filter and root samples (0.4 g) with an adapted phenol–chloroform based extraction protocol[74] with three rounds of mechanical distribution[75] via bead beating (30 s at 6.5 m s$^{−1}$, FastPrep-24 bead beater; MP Biomedical, Heidelberg, Germany) as described in[76].

The rhizoplane was sampled using a modification of a previously established protocol[71]. As described above, samples consisted of nucleic acids extracted from the rhizoplane slurry and filter; eluted nucleic acids then were mixed in a 1:1 ratio and used as a combined rhizoplane sample for further analysis. Nucleic acids were purified using the OneStep$^{TM}$PCR Inhibitor Removal Kit (Zymo Research, Irvine, CA, USA), and DNA was quantified using the Quant-iT$^{TM}$ PicoGreen®dsDNA assay (ThermoFisher Scientific, Waltham, MA, USA, both according to the manufacturer's protocol.

RNA was purified using Turbo DNAse (ThermoFisher Scientific, Waltham, MA, USA). Briefly, approximately 1–3 μg of DNA was digested with 2 μl Turbo DNAse (ThermoFisher Scientific, Waltham, MA, USA) and concentrations using the GeneJET RNA Cleanup and Concentration Micro Kit (Thermo Fischer, Waltham, Massachusetts, US), following the manufacturer's protocol. Successful DNA digestion was confirmed by no PCR amplification after 30 cycles with the general 515F-mod (5′ GTG YCA GCM GCC GCG GTA A 3′) and 806-mod (5′ GGA CTA CNV GGG TWT CTA AT 3′) primers[77,78] using the purified RNA as a template. For cDNA synthesis, 50–200 ng of RNA sample were used in a reaction containing random hexamers and SuperScript IV reverse transcriptase (ThermoFisher Scientific, Waltham, MA, USA) according to the manufacturer's protocol.

## Sample preparation for 16S rRNA gene and *nifH* gene amplification via MiSeq Illumina sequencing

To amplify the 16S rRNA gene and the *nifH* gene from extracted samples, gene-specific primers containing a universal 5′-end head sequence were used in a two-step PCR barcoding approach[79]. In short, amplification of the target region was done in triplicates in 20 μl first-step PCR reactions. Primer pairs H-515F-mod and H-806-mod (515 F: 5′-H-GTG YCA GCM GCC GCG GTA A-3′; 806 R: 5′-H-GGA CTA CNV GGG TWT CTA AT-3′)[77,78] targeting the V3 and V4 regions of the 16S rRNA gene in bacteria and archaea were used. For *nifH* functional gene amplification, the primer set Ueda19F (5′ GCI WTY TAY GGI AAR GGI GG 3′)[80] and R6 (5′ GCC ATC ATY TCI CCI GA 3′)[81] based on previously systematic evaluations of the *nifH* primers[82].

The 16S rRNA gene was amplified in a first-step 20 μl PCR reaction containing 2 μl of 10× Dream Taq Buffer, 2 μl 2 mM dNTP mix, 0.08 μl BSA (0.08 μg μl$^{−1}$), 0.2 μl of 1.25U DreamTaq Green DNA Polymerase, 0.5 μl of each 10 μM primer, and 1 μl DNA template (*ca.* 10–20 ng per reaction). After an initial denaturation step at 94 °C for 4 min, 22 cycles of 30 s denaturation at 94 °C, 45 s annealing at 52 °C and 45 s elongation at 72 °C followed. A final elongation step at 72 °C for 10 min was included. The amplification was done in three technical replicates.

The 20 μl PCR reaction for *nifH* gene amplification contained 2 μl of 10× Dream Taq Buffer, 2 μl 2 mM dNTP mix, 0.16 μl BSA (0.16 μg μl$^{−1}$), 0.2 μl of 1.25 U DreamTaq Green Polymerase, 2.6 μl of each 10 μM primer, and 1 μl DNA template (*ca.* 10–20 ng per reaction). After an initial denaturation step at 94 °C for 4 min, 32 cycles of 30 s denaturation at 94 °C, 45 s annealing at 52 °C and 30 s elongation at 72 °C followed. A final elongation step at 72 °C for 10 min was included. The amplification was done in three technical replicates.

After pooling the triplicate reactions and purification using the ZR-96 Clean-up kit$^{TM}$ (Zymo Research, Irvine, USA), the purified product was used as a template in a 50 μl s step PCR (8 cycles), generating the sample-specific barcoding. The second step PCR reactions contained 5 μl 10× DreamTaq Buffer, 5 μl 2 mM dNTP mix, 0.2 μl BSA (0.08 μg μl$^{−1}$), 0.25 μl of 1.25 U DreamTaq Green Polymerase, 4 μl of 10 μM head-barcode primer and 3 μl DNA template (16S rRNA gene) or 3–5 μl DNA/cDNA template

(*nifH* gene). After an initial denaturation of 4 min at 94 °C, 8 cycles comprising a 30 s denaturation at 94 °C, 30 s annealing at 52 °C and 45 s elongation at 72 °C followed. The single elongation step at 72 °C for 10 min was included.

Samples were further purified with the ZR-96 Clean-up kit™. For *nifH* samples with multiple amplifications, the band of interest was extracted from an agarose gel and purified using the QIAquick Gel extraction kit (QIAGEN, California, USA). Final purified products were pooled in equimolar amounts of $20 \times 10^{-9}$ copies per sample library. Sequencing was performed at Microsynth AG (Balgach, Switzerland). The library was prepared by adapter ligation and PCR using the TruSeq Nano DNA Library Prep Kit (Illumina, United States) according to the TruSeq nano protocol (Illumina, FC-121-4003), but excluding the fragmentation step. The MiSeq was run in the $2 \times 300$ cycle configuration using the MiSeq Reagent kit v3 (Illumina).

### Data analysis
The raw amplicon reads were merged using BBmerge v.37.61 with strict setting (exact match required) and a minimum overlap of 50 bp after clipping 3′-prime ends with quality scores below 20. - Exact sequence variants (ASVs) were generated and grouped into percentage-identity-independent operational taxonomic units (OTUs) with the intention to analyze the data at a meaningful taxonomic level roughly corresponding to species level. More specifically, ASVs were generated based on the entire dataset (pool=TRUE settings) using DADA2[83] with otherwise standard settings and then grouped into OTUs with SWARM2[84] in fastidious mode with a limit of a large swarm for grafting set at 20. The taxonomic assignment to OTU-centroids by the last common ancestor (LCA) algorithm was done using rRNA-secondary-structure-aware SINA aligner v.1.2.11[85] and the SILVA SSU138 database[86]. ASVs generated from *nifH* reads were translated using Framebot[87] and filtered using HMMscan against nifH, ChlL, and bchX HMMs through the bioinformatic pipeline NifMAP[82]. *NifH* ASVs were then taxonomically classified using the best hit returned from Diamond BLASTP[88] against the NCBI non-redundant database and grouped into OTUs as described above.

### qPCR quantifications of *nifH* gene copy numbers
The *nifH* gene copy numbers were quantified from NA extracts of bulk and rhizosphere samples of grasses and herbs from unfertilized, N-, P-, and NPK-fertilized plots. qPCR assays were performed in triplicates. The 20 µl qPCR reactions contained 10 µl of 1× iQ SYBR Green Supermix (BioRad, CA, United States), 0.2 µl nuclease-free water, 0.2 µl BSA (0.08 µg µl⁻¹), 2.8 µl of each primer (Ueda19F/R6) and 1–9 ng of template. The *nifH* standard was generated using a cloned fragment of the *nifH* gene from *Didymococcus colitermitum* TAV2 (ATCC BAA-2264)[89] ranging from $6.5 \times 10^5$ to $6.5 \times 10^0$ copies. The assays were run on a Bio-Rad C1000 CFX96 Real-Time PCR system (Bio-Rad, Hercules, CA, USA) with the following program: 94° for 4 min, followed by 45 cycles of 30 s at 95 °C, 45 s at 52 °C, 30 s at 72 °C, 10 s at 78 °C. Melting curves were generated between 55 °C and 95 °C. Data were processed and analyzed using the CFX Manager software (Bio-Rad, Hercules, CA, USA).

### Isotope ratio mass spectrometry (IRMS) measurements
NA was extracted from 0.4 g soil of the $^{15}N_2$ incubation experiment according to the protocol described above. $^{15}N/^{14}N$ isotope ratios in DNA extracts were determined by elemental analysis-isotope ratio mass spectrometry (EA-IRMS: EA 1110; CE Instruments, Milan, Italy, coupled to a Finnigan MAT DeltaPlus IRMS; Thermo Finnigan, Bremen, Germany.

### Statistics and reproducibility
Exact numbers of investigated samples for *nifH* and 16S rRNA gene sequencing (all microenvironments associated with grasses and herbs across the fertilization treatments), for sequencing, expressed *nifH* genes from cDNA samples of grasses in the unfertilized treatment, and for N₂ fixation potential analysis are listed in Supplementary Table 8. Data analysis and statistics were done in R (version 4.2.2)[90]. For sequence analysis, packages

vegan[91] and phyloseq[92] were used. Figures were created using the package ggplot2[93].

To remove low abundant OTUs, all OTUs (from both *nifH* and 16S rRNA gene) with a total read number of less than 0.1% relative abundance in at least 1% of the samples were removed in a first filtering step using the function genefilter_sample of the package phyloseq[92].

Samples were rarefied to even sampling depth for beta diversity analysis and Shannon diversity index calculations (*nifH*: 1262 reads, 16S rRNA gene: 1494 reads). To compare the cDNA data to the DNA data, only DNA samples for which a respective cDNA sample existed were used (rarefied to 1047 reads), some cDNA samples did not result in reads.

On the Shannon index, ANOVA was followed by the TukeyHSD post-hoc test whenever the assumption of normality on the residuals was met[94], otherwise a non-parametric Kruskal-Wallis rank sum test was applied combined with the Dunn post-hoc test for multiple comparisons of the package FSA[95]. To test the effects of fertilization, plant type, plant species, or microenvironment on the Bray–Curtis dissimilarity matrix, permutational multivariate analysis of variance (PERMANOVA) was performed with 9999 permutations using the adonis2 function in the R package vegan in combination with the post hoc test pairwiseAdonis. *P*-values adjusted to Benjamini–Hochberg corrections for multiple testing[96] are displayed. To perform an analysis of multivariate homogeneity (PERMDIST), the function betadisper from the R package vegan was used with bias.adjust=T argument[97]. ANOVA was used to compare among-group differences in the distance from observation to their group centroid. Differential abundance analysis was done using the R package DESeq2[98] to detect differential OTUs (log2 fold change) between pairwise microhabitat comparisons among fertilization treatments or between pairwise fertilization comparisons among microhabitats. A pseudocount of 1 was added to the count data prior to the analysis, to avoid applying a log on 0 values. Based on the DESeq2 analysis, OTUs were considered significantly differentially abundant with a *p*-value lower than 0.05 (Benjamini–Hochberg adjustment).

### Reporting summary
Further information on research design is available in the Nature Portfolio Reporting Summary linked to this article.

### Data availability
The raw sequence data were deposited in the NCBI Short Read Archive under study accession number PRJNA961667. DeSeq2 results of differentially abundant OTUs can be found in Supplementary Data 1, sheets 1.1–1.3. Source data underlying Fig. 3 can be found in Supplementary Data 1, sheet 1.4, and source data underlying Fig. 6 can be found in Supplementary Data 1, sheet 1.5. OTU tables, Metadata, and Taxonomy files used in the analysis are deposited on figshare and available at https://doi.org/10.6084/m9.figshare.25992034

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

## Acknowledgements

This research was funded in whole or in part by the Austrian Science Fund FWF (Grant-DOI: 10.55776/P25700 and 10.55776/W1257). For open access purposes, the author has applied a CC BY public copyright license to any author-accepted manuscript version arising from this submission. We thank the Division of Computational Systems Biology and the University of Vienna for providing and maintaining excellent computation resources (Vienna Life Science Compute Cluster). We would like to thank Margarete Watzka for her assistance with the IRMS measurements and members of the Woebken Group, especially Maximilian Nepel and Stefanie Imminger, for their fruitful discussions about the data. We also would like to thank the team of the Austrian Research and Education Center Raumberg-Gumpenstein (AREC) for managing the experimental site and for their support during the sampling campaigns.

## Author contributions

D.W. and R.A. designed the study. M.D., D.W., R.A., C.P., and A.T.G. collected and prepared samples for analysis. E.P. and A.S. provided metadata of the studied soils and access to the sampling site. M.D. and C.P. carried out the extractions and PCRs, and M.D. and A.T.G. performed qPCR assays. M.D. and D.R. performed the activity assays. B.H. helped in the processing of the sequences, C.W.H. adapted the NifMAP pipeline for analyzing DADA2-processed amplicon reads grouped in SWARM2-OTUs, and M.D. performed sequence and data analysis. M.D. wrote the paper with the support of D.W. and S.A.E. All co-authors contributed to revisions of the paper.

## Competing interests

The authors declare no competing interests.
