## [Peer review file · Communications Biology]

Reviewers' comments:

Reviewer #1 (Remarks to the Author):

The manuscript 'Plant roots affect free-living diazotroph communities in temperate grassland soils despite decades of fertilization' by Dietrich et al. investigated the effects of long term fertilization on the soil bacterial and diazotrophic communities in the bulk soil, rhizosphere, rhizoplane and root samples from four temperate grassland species.

The concept of the study was very interesting. I found particularly novel (and important) the finding that P-fertilized soils exhibit increased N₂ fixation. However, the way in which the study was conducted and the reporting of the results are quite serious drawbacks.

- 1) L170: I would advise listing the hypotheses at the end of the introduction section.
- 2) L130 – 138: Totally unclear what you are analysing here – the three root compartments combined?
- 3) L140 –L142: What do you mean by analyzed individually? Why would they not be analysed separately? I don't see a justification for doing otherwise.
- 4) L345: Relating to Figure 7, I like the figure, but unclear what the scale for the arrow size is supposed to represent.
- 5) L452: The same treatments and amounts have been added since 1960? This is not clear.
- 6) L470-476: It is totally unclear why three of the fertilizer treatments were sampled in 2013 and 1 in 2018? The authors say they tested for an effect of time, but this is totally unclear how it was done. If samples were available over time, why not just sample all in 2018? Also, when exactly were the samples taken in each year? When is 'before the second cut'?
- 7) L493: For the bulk soil samples, why were 'four to five' cores taken? Why not the same number of cores per plot? Do the authors think these samples are comparable as a result?
- 8) L511 – 513: Why did the authors vary the amount of material to extract DNA from?
- 9) L516-517: It is difficult to see how the extract method utilized could result in a representative 'rhizoplane' sample.
- 10) L547: '10-20 ng per reaction' why was the DNA not standardized to the same concentration, so all PCR reactions could have the same amount? Also unclear why the authors first created ASVs, then OTUs?
- 11) L622: Unclear why there were differences in sample size.
- 12) Results section in general: A distinct lack of any reported values throughout the results section, making it difficult for the reader to get a grasp for the effect size for the measured treatments. For example, L208: How much more abundant?
- 13) Discussion section: Well written, but I feel it could be improved by being broken up into different sub-headings.

Reviewer #2 (Remarks to the Author):

This manuscript reports the effects of long-term fertilization and vicinity to the roots of 4 plant species on diazotrophic and general bacterial community composition, abundance and potential activity. Amplicon sequencing of the nifH and 16S rRNA genes was employed to provide information about the structure of these communities, and, using cDNA, to investigate

active diazotrophs in plant-associated microenvironments of two grass species. The abundance of diazotrophs in the rhizosphere and bulk soil was investigated using qPCR. Nitrogen fixation potential was also tested in the presence and absence of root exudates.

General comments:

This is a nice study, the diazotroph community shifts across a gradient from bulk soil to plant roots are clear and novel. The application of amplicon sequencing to compare diazotroph and general microbial communities, and the assessment of the active diazotroph community composition are nice aspects of this work. I think it is worthwhile to publish these results; however, I have a few concerns and recommendations.

First, I am sceptical about the use of data from the NPK sites, where samples were collected five years after the other plots. Quite significant changes in soil microbial community composition can occur over time resulting from seasonal changes, year-to-year changes in precipitation and temperature, and/or disturbances. Because these data were merged with the 2013 data set, the NPK samples are compared to the unfertilized control from 5 years earlier. Although the authors performed a PERMANOVA to look for differences in the unfertilized, N and P plots in 2013 vs. 2018, few details about the samples used in this test are included (l. 473-476). How many replicates were taken? Were these e.g. only bulk soil samples, or were samples taken from all microenvironments? Were the samples taken in the same month/season as 2013? This also raises a question about the potential nitrogen fixation activity – here there is data from NPK plots; however, this must be performed using fresh soil. Were all samples for this assay collected in 2018, or were the N, P, and unfertilized samples assayed in 2013, and only the NPK later? Without additional information about the sampling, it is difficult to assess if it was valid for these data to have been included as they were in statistical models.

The fertilization and plant effects on the diazotrophic community are quite clear when looking at your data visualization and statistical results. This didn't come across so easily in the text of the Results section – I only really got what was going on when I got to the end of the section. I would recommend starting with the main model with plant species, fertilization and microenvironment, and only discussing differences between plant type later. I would also recommend reducing the amount of discussion around plant type and the many pairwise comparisons – e.g. choose a few that highlight particularly novel or interesting results (or are required to justify decisions).

Finally, a small comment on the plant-associated microenvironments: some background information in the introduction, e.g. differences in the soil physico-chemical environment and microbial communities across them and/or compared to bulk soil would help support your hypotheses, particularly because these results are a highlight of this manuscript.

Specific comments:

l. 39-40: the effects of fertilization on the diazotrophic community is fairly well resolved; perhaps rephrase to make this clear

l. 59-61: I don't know that I agree that the importance of free-living diazotrophs is underestimated. To provide context for the reader, perhaps replace this sentence with an estimate of the amount/percentage of N₂ fixation that they are responsible for.

l. 96-97: “Furthermore, the effects of different native plant species (grown in the same treatments in temperate grassland)...” – what does this mean? Sounds like it’s referring to an experimental design, but is also formulated as a general statement

l. 139 – 142: It would be more clear to include “pairwise” in the text here for clarity.

Figure 1: nice clear pattern in the ordination across microenvironments here. I would recommend merging the grass and herb ordinations (at least for the fertilization results).

l. 234-235: It would be helpful for the reader to add a sentence at the beginning of the section stating which samples were used for this analysis (all microenvironments for the two grass species in the unfertilized plots).

Figure 5. Panels c and d would be easier to interpret if these were shown as percentages, or at least if the total number of OTUs is given on each panel

l. 321-322: This section would work well at the beginning of the Results section.

Figure 7: I have mixed feelings about this figure. It is visually appealing and makes a nice summary of the overall results, but it has the feeling of a graphical abstract. I think it would work better with a quantitative measure included (e.g. the amount of explained variance), and with the arrows differing only in length (some of them also differ in width).

l. 423: is the unfertilized soil “low nutrient”? Only the soil type, but not information about the soil physico-chemical conditions is provided in the M&M.

l. 431-432: I would rephrase this to highlight that your experimental design was unique in investigating communities shaped over decades of fertilization treatment rather than shorter-term fertilization treatments often applied

l. 436-438: this is not only relevant in long-term fertilization experiments. Also, I don’t know that the work highlights that plant-associated microenvironments are important to study in general (this would double the number of samples in many cases), but that it would be informative in certain contexts

l. 443-444: I don’t know that the conclusion necessarily follows from the premise here.

Furthermore it would be nice to end with a statement that discusses the impact of your work.

l. 452: some details about the physico-chemical properties of the soil would be nice here (at least for the unfertilized plots)

l. 473-476: please see my general comment about the additional details requested here

Reviewers' comments:

Reviewer #1 (Remarks to the Author):

I appreciate the considered response of the authors to my original comments, I think the authors have clarified my original concerns.

I think this manuscript is now suitable for publication, after the following small corrections:

Figures: 2D stress values should be added to add NMDS plots

L345: Should read 'then' not 'than'

Reviewer #2 (Remarks to the Author):

In the revised manuscript many of my concerns have been addressed, particularly regarding context and clarity. The Introduction section revisions are nice; the table with the soil edaphic conditions helps to provide important ecological context. The Methods section is also now more clear.

However, I am still not convinced by the inclusion of the NPK samples. The authors have now included a supplementary figure and statistical analysis with some support for the claim that there were not major shifts in community composition between 2014 and 2018 within the unfertilized, N and P bulk soil. However, I do not believe that this justifies unbridled inclusion of the NPK samples in analyses throughout the manuscript.

Temporal shifts in soil microbial communities are well-documented, and sampling, even from the same plot, at different time points means sampling different microbial communities. The unfertilized, N and P samples represent microbial communities differing in long-term fertilization treatments, but that experienced the same conditions prior to sampling (e.g. rainfall, temperature, time since last frost). The NPK samples on the other hand differ in conditions prior to sampling in addition to long-term fertilization treatment. With the data presented here it is impossible to disentangle the effects of fertilization treatment from the effects of time when comparing the NPK samples to the other fertilization treatments.

In my opinion, the difference in sampling time must somehow be included in the text and figures displaying all fertilization treatments so that the results can be interpreted appropriately. Particularly given that the M&M is at the end of the manuscript, this information should appear earlier in the text, at the latest at the beginning of the section "N fertilization reduced diazotroph diversity and abundance and selected for distinct community members in bulk soils". Figure 5 relates to bulk soil microbial communities – is all the data shown here from the 2018 sampling campaign? This should be stated in the Figure caption. If 2014 and 2018 data are mixed here, then I would suggest at least using a different point shape in the NMDS plots to indicate this to the reader, and explaining the data used in the figure caption. Please also reference the combinatorial analysis performed excluding the NPK samples in the section "Combinatorial analysis reveals increasing effect of plant on diazotroph and general microbial community structure with vicinity to the root".

Microbial abundance is likely to be more sensitive to recent abiotic conditions (e.g. rainfall/soil moisture) than community structure. Therefore, I think it would be appropriate to separate the NPK sample into a separate facet in Figure 3 and Supplementary Figure 3. In the main manuscript text, you only report statistical differences between microhabitats (i.e. within and not across fertilization treatments; this is fine), however, plotting the treatments together inherently encourages the reader to compare fertilization treatments. There it appears that the NPK samples have the lowest abundance of diazotrophs. However, the reduced abundance of diazotrophs in the NPK samples may well be alternately explained by e.g. weather conditions prior to sampling.

In Supplementary Figure 3, the diazotroph abundances of the NPK treatment are statistically compared to the other fertilization treatments; I do not think this is justified. I would recommend

separating the NPK treatment into a separate panel and only performing the statistical analysis to compare field treatments between the unfertilized, N and P treatments. Which data are included in the Kruskal-Wallis test - are these comparisons between pooled data from each field treatment? What type of adjustment was used to correct for multiple comparisons in the Dunn posthoc tests? This information should be indicated in the figure caption or the text.

Some additional minor points to address: (1) Please include stress values in the NMDS plots to allow assessment of the goodness of fit. (2) Many numbers, including p-values, are written strangely. Presumably this is a misuse of 'E notation' (i.e. what R spits out), where 'e' stands for exponent, or 'ten to the power of'. The exponent then shouldn't be written as a superscript. As it reads now it looks like 'the number e (base of the natural log) raised to the power of'.

Reviewer #1 (Remarks to the Author):

I appreciate the considered response of the authors to my original comments, I think the authors have clarified my original concerns.

I think this manuscript is now suitable for publication, after the following small corrections:

Figures: 2D stress values should be added to add NMDS plots

Stress values were added to the figure legends in all NMDS plots in the manuscript.

L345: Should read 'then' not 'than'

Thank you for the suggestion, but as we are making a comparison, we believe “than” is correct.

Reviewer #2 (Remarks to the Author):

In the revised manuscript many of my concerns have been addressed, particularly regarding context and clarity. The Introduction section revisions are nice; the table with the soil edaphic conditions helps to provides important ecological context. The Methods section is also now more clear.

However, I am still not convinced by the inclusion of the NPK samples. The authors have now included a supplementary figure and statistical analysis with some support for the claim that there were not major shifts in community composition between 2014 and 2018 within the unfertilized, N and P bulk soil. However, I do not believe that this justifies unbridled inclusion of the NPK samples in analyses throughout the manuscript.

Temporal shifts in soil microbial communities are well-documented, and sampling, even from the same plot, at different time points means sampling different microbial communities. The unfertilized, N and P samples represent microbial communities differing in long-term fertilization treatments, but that experienced the same conditions prior to sampling (e.g. rainfall, temperature, time since last frost). The NPK samples on the other hand differ in conditions prior to sampling in addition to long-term fertilization treatment. With the data presented here it is impossible to disentangle the effects of fertilization treatment from the effects of time when comparing the NPK samples to the other fertilization treatments.

In my opinion, the difference in sampling time must somehow be included in the text and figures displaying all fertilization treatments so that the results can be interpreted appropriately. Particularly given that the M&M is at the end of the manuscript, this information should appear earlier in the text, at the latest at the beginning of the section "N fertilization reduced diazotroph diversity and abundance and selected for distinct community members in bulk soils".

We added a sentence describing the different sampling years in the Results section, even earlier than suggested by the reviewer. We now mention it in the last part of the “Plant root vicinity selected for distinct, less diverse diazotroph communities” (Page 10, lines 234-237), when the qPCR data are presented.

Figure 5 relates to bulk soil microbial communities – is all the data shown here from the 2018 sampling campaign? This should be stated in the Figure caption. If 2014 and 2018 data are mixed here, then I would suggest at least using a different point shape in the NMDS plots to indicate this to the reader, and explaining the data used in the figure caption.

To clarify the sampling years in Figure 5, we added the dates for the respective fertilization treatment to the color legend in this figure. In addition, the sampling years for the respective fertilization treatment have been further articulated in the legend.

Please also reference the combinatorial analysis performed excluding the NPK samples in the section "Combinatorial analysis reveals increasing effect of plant on diazotroph and general microbial community structure with vicinity to the root".

The combinatorial analysis performed excluding the NPK samples was added in the suggested section, Page 16, line 353-355.

Microbial abundance is likely to be more sensitive to recent abiotic conditions (e.g. rainfall/soil moisture) than community structure. Therefore, I think it would be appropriate to separate the NPK sample into a separate facet in Figure 3 and Supplementary Figure 3. In the main manuscript text, you only report statistical differences between microhabitats (i.e. within and not across fertilization treatments; this is fine), however, plotting the treatments together inherently encourages the reader to compare fertilization treatments. There it appears that the NPK samples have the lowest abundance of diazotrophs. However, the reduced abundance of diazotrophs in the NPK samples may well be alternately explained by e.g. weather conditions prior to sampling.

In Figure 3 and Supplementary Figure 3, we have added the facets as suggested by the reviewer and stated the different sampling years in the figure legend. In addition, we mentioned the different sampling years in the text of this section (Page 10, lines 234-237).

Figure 3:

Supplementary Figure 3:

In Supplementary Figure 3, the diazotroph abundances of the NPK treatment are statistically compared to the other fertilization treatments; I do not think this is justified. I would recommend separating the NPK treatment into a separate panel and only performing the statistical analysis to compare field treatments between the unfertilized, N and P treatments. Which data are included in the Kruskal-Wallis test - are these comparisons between pooled data from each field treatment? What type of adjustment was used to correct for multiple comparisons in the Dunn posthoc tests? This information should be indicated in the figure caption or the text.

We have moved the NPK treatment into a separate facet as suggested by the reviewer. The statistics were adapted to only compare field treatments between unfertilized, N and P treatments. Additional information on the statistical tests, which were stated in the M&M section, are now also added in the figure caption for clarity (Page 10, lines 243-244).

Some additional minor points to address: (1) Please include stress values in the NMDS plots to allow assessment of the goodness of fit. (2) Many numbers, including p-values, are written strangely. Presumably this is a misuse of 'E notation' (i.e. what R spits out), where 'e' stands for exponent, or 'ten to the power of'. The exponent then shouldn't be written as a superscript. As it reads now it looks like 'the number e (base of the natural log) raised to the power of'.

- (1) Stress values were added in the figure legends to all NMDS plots throughout the manuscript.
- (2) The "e notation" was adapted throughout the manuscript.